# Finding Support Examples for In-Context Learning

**Xiaonan Li, Xipeng Qiu**
School of Computer Science, Fudan University
Shanghai Key Laboratory of Intelligent Information Processing, Fudan University
{lixn20, xpqiu}@fudan.edu.cn

## Abstract

In-context learning is a new learning paradigm where a language model observes a few examples and then directly outputs the test input's prediction. Previous works have shown that it is sensitive to the provided examples and randomly sampled examples probably cause inferior performance. In this paper, we propose finding "support examples" for in-context learning: Given a training dataset, it aims to select one permutation of a few examples, which can well characterize the task for in-context learning and thus lead to superior performance. Although for traditional gradient-based training, there are extensive methods to find a coreset from the entire dataset, they struggle to identify important in-context examples, because in-context learning occurs in the language model's forward process without gradients or parameter updates and thus has a significant discrepancy with traditional training. Additionally, the strong dependency among in-context examples makes it an NP-hard combinatorial optimization problem and enumerating all permutations is infeasible. Hence we propose **LENS**, a fiLter-thEN-Search method to tackle this challenge in two stages: First we filter the dataset to obtain informative in-context examples individually. Specifically, we propose a novel metric, InfoScore, to evaluate the example's in-context informativeness based on the language model's feedback, and further propose a progressive filtering process to filter out uninformative examples. Then we propose diversity-guided example search which iteratively refines and evaluates the selected example permutations, to find examples that fully depict the task. The experimental results show that LENS significantly outperforms a wide range of baselines.

## 1 Introduction

In-Context Learning (ICL) is a new paradigm using the language model (LM) to perform many NLP tasks (Brown et al., 2020; Dong et al., 2022; Zhao et al., 2023). In ICL, by conditioning on a few train-

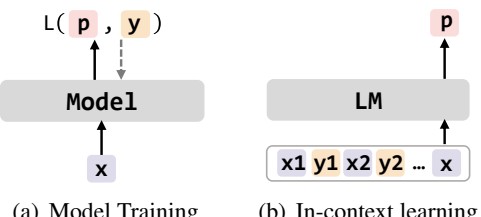

(a) Model Training      (b) In-context learning

Figure 1: In-context learning and model training learn in different ways , where ↑ and ↓ are forward and backward processes, respectively, and L(·,·) is the loss function.

ing examples, LM can directly output the prediction of a given test input without parameter updates. Restricted by LM's max input length, it is typical to randomly sample a small set of examples from the entire dataset for in-context learning (Brown et al., 2020; Zhang et al., 2022a). However, in-context learning is sensitive to the provided examples and randomly sampled in-context examples show significant instability and probably cause inferior performance (Lu et al., 2022; Chang and Jia, 2023). In this paper, we propose to select a small list of examples that are informative and representative for the entire dataset as in-context examples. Inspired by the traditional machine learning method, Support Vector Machine (SVM) (Cortes and Vapnik, 1995), where a few support vectors are closest to the decision boundary and provide crucial discriminative information for SVM, we name the selected examples for ICL as **support examples** since they provide crucial task information for the LM and their quantity is usually limited, too.

There is a similar problem in traditional gradient-based deep learning like fine-tuning (Devlin et al., 2019), typically called Coreset Selection (Guo et al., 2022), which aims to select a set of representative training examples for the dataset to benefit many downstream scenarios like data-efficient learning (Adadi, 2021), active learning (Ren et al., 2022), neural architecture search (Shim et al., 2021), etc. However, it is challenging for these coreset selection methods to select important in-

| Examples | Acc |
|---|---|
| The casting of Raymond J. Barry as the 'assassin' greatly enhances the quality ... It was [great] | 82.0 |
| At some point , all this visual trickery stops being clever ... It was [terrible] | 85.3 |
| 1. The casting of Raymond ... It was [great] 2. At some point , all this visual trickery... It was [terrible] | 56.0 |
| 1. At some point , all this visual trickery ... It was [great] 2. The casting of Raymond ... It was [terrible] | 74.4 |

Table 1: The case study of examples' combinatorial dependency on SST-2, where "it was [great/terrible]" is the template. Although two examples bring good performance separately, combining them instead hurts performance.

context examples because there is a significant discrepancy between traditional training and ICL. As shown in Figure 1, the "learning" paradigms of model training and ICL are highly different. Traditional training depends on back-propagation's gradients to update parameters while ICL occurs in LM's forward process without gradients and parameter updates. Existing coreset selection methods are always coupled with the training procedure, i.e., they usually depend on gradients or run with the training procedure. For example, Paul et al. (2021) select informative examples by their gradients' norm. Toneva et al. (2019) evaluate each example's importance by counting how many times it is forgotten, i.e., the example is misclassified after being correctly classified in the previous epoch. Additionally, coreset selection methods mainly depend on the example's gradients as the feature of example selection (Mirzasoleiman et al., 2020; Killamsetty et al., 2021a,b; Guo et al., 2022). However, LM performs ICL through inference, which does not rely on gradients or parameter updates. Hence, the gap between gradient-based training and ICL makes these methods struggle to effectively select informative examples for in-context learning.

Another challenge is the strong dependency among in-context examples. Previous work (Lu et al., 2022) shows that even the same example set with different orderings can result in drastically different performance from random-guess level to state-of-the-art. Here we also conduct an additional case study to shed light on examples' combinatorial dependency in Table 1. We see that compared with two examples' individual performance, combining them instead significantly hurts the performance. To cope with examples' dependency, a straightforward method is to enumerate all possible examples'

combinations and verify their performance. However, it will lead to combinatorial explosion and thus is infeasible.

To tackle these challenges, we propose **LENS**, a fi**L**ter-th**EN**-**S**earch that finds support examples in two stages: in the first stage, we filter the dataset to obtain informative in-context examples individually. Specifically, we propose InfoScore to evaluate the example's in-context informativeness based on the LM's feedback, and further propose a progressive filtering process to filter out uninformative examples; In the second stage, we propose a diversity-guided example search method that iteratively refines and evaluates the selected examples to find support examples that can fully depict the task. We summarize our contributions as follows:

- To the best of our knowledge, we are the first to define the support examples selection problem for in-context learning and introduce a novel filter-then-search method to tackle it.

- We conduct experiments on various text classification datasets and compare our method with a wide range of baselines. Experimental results demonstrate that our method significantly outperforms baselines and previous coreset selection methods bring marginal improvements over the random baseline, which shows the necessity of ICL-specific designing for finding support examples.

- We conduct further analyses on support examples and find that they exhibit different trends from random examples in many aspects, which can shed light on the principle of them and ICL. We provide the following **key takeaways**: 1. Support examples are less sensitive to the order compared with random examples (Lu et al., 2022). 2. Ground truth labels matter for support examples, while the previous study (Min et al., 2022b) show that they are not important for randomly sampled examples. 3. One LM's support examples can be well transferred to other LMs with different sizes and pre-training corpora and keep the superiority over random examples.

- We provide comprehensive empirical results of previous coreset selection methods on ICL, which has not been explored. We release the implementation of our method and baselines to facilitate future research[1].

---

[1]https://github.com/LeeSureman/ICL_Support_Example

## 2 Background: In-Context Learning

In this section, we introduce the definition of in-context learning. We focus on text classification's in-context learning using the causal language model (Radford et al., 2018). Given a language model $G$, $n$ examples $\{x_i, y_i\}_{i=1}^n$ and a test input $x_{test}$, the prediction of $x_{test}$ is generated as:

$$\arg\max_{y \in \mathcal{Y}} p_G(y|x_1 \oplus y_1 \cdots x_n \oplus y_n \oplus x_{test}), \quad (1)$$

where $\mathcal{Y}$ is the label space and $\oplus$ is the concatenation operation. To deal with classification tasks, the original label is often mapped to word or words in $G$'s vocabulary. For example, the positive/negative label in a binary sentiment classification can be mapped to "*great*"/"*terrible*". For simplicity, we omit the verbalizer, special tokens and prompting templates in Eq (1).

As Eq.(1) shows, $G$ receives the task's supervision only from the concatenated $\{x_i, y_i\}_{i=1}^n$ and directly output the prediction of $x_{test}$. Typically, $n$ is limited by the max input length of $G$, so it is typical for researchers to randomly sample a small set of samples from the entire dataset $\mathcal{D}$ (Brown et al., 2020; Zhang et al., 2022a). However, ICL is sensitive to the provided examples and random in-context examples show significant instability and probably cause inferior performance(Lu et al., 2022; Chen et al., 2022). In this paper, we focus on selecting a small list of support examples that are informative for the task and performant for in-context learning, from the entire dataset $\mathcal{D}$.

## 3 Method

The strong dependency among in-context examples makes selecting support examples essentially an NP-hard combinatorial optimization problem. Enumerating all combinations and evaluating them is infeasible due to the combinatorial explosion. In this section, we propose **LENS**, a fiLter-thEN-Search method to find support examples: 1. we first filter the training dataset to obtain informative examples individually, 2. then we search the example permutation that fully depicts the task from them. In this paper, we instantiate the two stages as a novel example metric with progressive filtering and diversity-guided example search, we leave the development of more powerful components as future work. We introduce these two stages below.

### 3.1 Informative Examples Filtering

In the first stage, we aim to find those informative examples individually. There are extensive

---

**Algorithm 1** Progressive Example Filtering

**Input:** Training set $\mathcal{D} = \{e_i\}_{i=1}^n$, language model $G$, desired candidate size $m$, progressive factor $\rho$, initial score data size $l$.
**Output:** Individually informative examples $\mathcal{D}'$
1: $\mathcal{D}' \leftarrow \mathcal{D}$
2: $S \leftarrow$ Randomly sample $l$ examples from $\mathcal{D}$.
3: **while** $|\mathcal{D}'| > m$ **do**
4:     **for** $e_i \sim \mathcal{D}'$ **do**
5:         $s(e_i) \leftarrow I(e_i, S)$
6:     **end for**
7:     **if** $|\mathcal{D}'|/\rho < m$ **then**
8:         $\mathcal{D}' \leftarrow$ the top-$m$ of $\mathcal{D}'$ using $\{s(e_i)\}_{i=1}^{|\mathcal{D}'|}$
9:         Break;
10:    **else**
11:        $\mathcal{D}' \leftarrow$ the top $\frac{1}{\rho}$ of $\mathcal{D}'$ using $\{s(e_i)\}_{i=1}^{|\mathcal{D}'|}$
12:    **end if**
13:    $S' \leftarrow$ Randomly sample $l * (\rho - 1)$ examples from $\mathcal{D}$
14:    $S \leftarrow S \cup S'$
15: **end while**
16: **return** $\mathcal{D}'$

---

methods to measure the example's importance for gradient-based training, like the example's gradient norm (Paul et al., 2021), loss value in the early training stage or the times of being forgotten (Toneva et al., 2019), etc. However, these methods struggle to identify important in-context examples since ICL is based on LM-inference without gradients and parameter updates. Here we propose *InfoScore (Informativeness Score)* to measure the individual in-context informativeness of one example $e = \{x, y\}$ for ICL based on LM's feedback as:

$$I(e, \mathcal{D}) = \sum_{e' \in \mathcal{D}} c(e, e') \quad (2)$$

$$c(e, e') = p_G(y'|x, y, x') - p_G(y'|x'), \quad (3)$$

where $e' = \{x', y'\}$, and $\mathcal{D}$ is the training dataset. Eq (3) is the gap between the probabilities of the ground truth $y'$ conditioned on $(e, x')$ and $(x')$, respectively. So it evaluates how informative $e$ is for the LM to correctly classify $x'$ and thus measures $e$'s contribution for $e'$ in ICL. Hence, $I(e, \mathcal{D})$, the sum of Eq (3) over $\mathcal{D}$, can evaluate the example's task-level in-context informativeness.

However, computing all examples' InfoScores over the entire dataset is quadratic in $|\mathcal{D}|$ and thus infeasible. We further propose a progressive filtering process to filter out uninformative examples progressively, where promising examples receive more computation while low-quality examples get less computation, shown in Algorithm 1.

We filter out uninformative examples in a progressive manner. We first sample a small set of examples from $\mathcal{D}$ as initial "score set" (line 2) to coarsely evaluate the InfoScore of each example

**Algorithm 2** Diversity-Guided Search

**Input:** Candidate examples $\mathcal{D}' = \{e_i\}_{i=1}^m$, candidates' feature $\{f(e_i)\}_{i=1}^m$, a small validation set $V$, iteration num $\mathcal{I}$, beam size $\mathcal{B}$, example substitution size $\mathcal{B}'$
**Output:** A performant examples' permutation.
1: $\mathcal{E} = \{E_i\}_{i=1}^{\mathcal{B}} \leftarrow$ initialize $\mathcal{B}$ examples' permutations
2: **for** $i$ in $1, 2 \cdots \mathcal{I}$ **do**
3:     $\mathcal{E}' \leftarrow \{\}$
4:     **for** $E$ in $\{E_j\}_{j=1}^{\mathcal{B}}$ **do**
5:         **for** $b$ in $1, 2 \cdots \mathcal{B}'$ **do**
6:             $e* \leftarrow$ Randomly sample an example from $E$
7:             $e^*_{new} \leftarrow \mathrm{argmax}_{e \in \mathcal{D}'}\, s(e, E - e^*)$
8:             $E^* \leftarrow$ Replace $e^*$ in $E$ with $e^*_{new}$
9:             $\mathcal{E}' \leftarrow \mathcal{E}' \cup \{E^*\}$
10:         **end for**
11:         **for** $b$ in $1, \cdots \mathcal{B} - \mathcal{B}'$ **do**
12:             $E^* \leftarrow$ Randomly shuffle $E$
13:             $\mathcal{E}' \leftarrow \mathcal{E}' \cup \{E^*\}$
14:         **end for**
15:         $\mathcal{E} \leftarrow$ Evaluate $\mathcal{E}'$ on $V$ and get the top-$\mathcal{B}$
16:     **end for**
17: **end for**
18: **return** The top-1 of $\mathcal{E}$

and filter the entire dataset to $1/\rho$ of its original size (line 5). At the following iteration, we proportionally expand the size of the score set to $\rho$ times by randomly sampling more examples from training set (line 13~15) and use it to calculate InfoScore of the remaining promising examples. As the score set is expanded, the subsequent InfoScore can be calculated in a more fine-grained way and better filter informative examples. Meanwhile, the uninformative examples are filtered out in the previous iteration, which helps save the computational cost. We repeat this procedure until a small set of examples is left.

Thus we achieve filtering examples with high in-context informativeness in the complexity of $O(N * \log_\rho N)$, where $N$ is the size of training set. In experiments, we set $\rho$ to $N^{\frac{1}{C}}$ to make it a linear complexity, where $C$ is a constant. According to the size of dataset, $\rho$ is usually set between 2 - 3.

### 3.2 Diversity-Guided Example Search

After filtering, we get individually informative examples $\mathcal{D}'$. Since the in-context examples have high combinatorial dependency (see Table 1), a straightforward method is to enumerate all possible combinations and evaluate them on a validataion set. However, although we have reduced the candidate examples by filtering, it is still impossible to evaluate all combinations. For example, if there are 50 examples retained after filtering and we want to find a combination of 8 examples from them, it can lead to $\mathbf{C}_{50}^8$ (about 536 million) combinations, let

alone considering the examples' orders.

Hence we propose diversity-guided example search to iteratively refine the example selection from filtered examples and obtain the support examples, as shown in Algorithm 2. It starts with a set of initial example permutations. At each iteration, we use the diversity of in-context examples to guide the update of the current candidate permutations. Specifically, for each candidate permutation $E = [e_i]_{i=1}^n$, we randomly select an example $e^*$ in $E$ and update it with the example $e^*_{new}$ as:

$$e^*_{new} = \mathrm{argmax}_{e \in \mathcal{D}'}\, s(e, E') \qquad (4)$$

$$s(e, E') = I(e, S) - \lambda \sum_{e' \in E'} \mathrm{sim}(f(e), f(e')), \quad (5)$$

where $E' = E - e^*$, $\lambda$ is pre-defined hyperparameter, $S = \{e_i^s\}_{i=1}^m$ is the final score set of the filtering stage. The subsequent term of $s(e, E')$ in Eq (5) corresponds to the diversity between $e$ and $E'$, and $f(\cdot)$ is the example's feature vector calculated as:

$$f(e) = [c(e, e_1^s), c(e, e_2^s) \cdots, c(e, e_{|S|}^s)], \qquad (6)$$

where $f(e)$ describes $e$'s contribution on $S$'s each example $e_i^s$ in ICL and thus directly encodes $e$'s in-context feature. If two examples' $f(\cdot)$ are similar, their effect on ICL can be redundant and we should avoid selecting both of them in one permutation. Note that $I(e, S)$ and each $c(e, e_i^s)$ in $f(e)$ are calculated in the filtering stage and can be reused.

With $s(e, E')$ and $f(e)$, the updated candidate permutations can be informative and diverse, and help the LM correctly predict various examples, which can better help find the support examples that fully depict the task in ICL. In this paper, we propose and verify a simple yet effective example ICL feature. We leave the development of more powerful ones as future work.

Since examples' order can significantly influence the performance (Lu et al., 2022; Kumar and Talukdar, 2021), we also update $E$ with different orders by randomly shuffling (line 10~13), which can reduce the risk of missing performant combinations of examples due to poor ordering. In order To explore the example search space more comprehensively and alleviate risk of the local-optimal example permutation, we consider the beam search (Jurafsky and Martin, 2009) here instead of greedy search. Specifically, we update each candidate example permutations by diversity-based example substitution and random shuffling for $B'$ and $B - B'$ times, respectively (line 5, 10). Then

we leverage a small validation set sampled from $(\mathcal{D} - \mathcal{D}')$ to evaluate them and keep the top-$\mathcal{B}$ permutations with best performance as next iteration's candidates. Through these, we can the mitigate issue of local-optimal example permutation, better iteratively refine and evaluate the candidate permutations with high informativeness and diversity in turn and obtain the examples that can fully depict the task.

To initialize example permutations $E$ with informativeness and diversity, we formulate it as discrete optimization that maximizes $\sum_{e \in E} s(e, E - e)$, which can be solved by the discrete optimization solver like CPLEX (Cplex, 2009).

## 4 Experiments

### 4.1 Experimental Settings

**Dataset** In this paper, we conduct experiments on eight text classification datasets across three task families, including **Sentiment Classification**: SST-2, SST-5 (Socher et al., 2013), Amazon (McAuley and Leskovec, 2013) and MR (Pang and Lee, 2005); **Subjectivity Classification**: Subj (Pang and Lee, 2004); **Topic Classification**: TREC (Voorhees and Tice, 2000), AGNews (Zhang et al., 2015) and DB-Pedia (Lehmann et al., 2015).

**Method Comparison** We mainly compare our proposed methods with the following baselines: **Random**: We randomly select examples from the training set; **Random & Validation**: We evaluate multiple sets of random examples on the validation set and select the best one. We consider Random & Validation under two settings whose computational cost is similar to our method: 1. the size of validation set is the same as ours at stage 2 (100) and the number of random example sets is the same as our searched and evaluated example permutations (640). 2. the size of validation set is larger, 1000, and the number of random example sets is 100. We also consider a wide range of **Coreset Selection** methods in gradient-based learning scenarios, according to the methodologies, they can be divided into multiple categories including: **Geometry-Based Method**: it assumes that data points that are close in the feature space have similar properties, including **Herding** (Chen et al., 2012) and **K-Center Greedy** (Sener and Savarese, 2018); **Uncertainty-Based Method**: it assumes examples with higher uncertainty can have a greater impact on the model and should be contained in coreset, including **Least Confidence**, **Entropy**, **Margin** (Coleman et al., 2020) and **CAL** (Mar-

gatina et al., 2021); **Error/Loss Based Method**: It assumes the example that contributes more to the error or loss during training is more important and should be included in coreset, including **Forgetting** (Toneva et al., 2019) and **GraNd** (Paul et al., 2021); **Gradient Matching Based Method**: Since deep models are usually trained by gradient descent, it tries to find a coreset whose gradients can imitate the entire dataset's gradients, including **CRAIG** (Mirzasoleiman et al., 2020) and **Grad-Match** (Killamsetty et al., 2021a); **Submodularity-Based Method**: Submodular functions (Iyer and Bilmes, 2013) naturally measure a subset's informativeness and diversity and can thus be powerful for coreset selection, including **Facility Location** and **Graph Cut** (Iyer and Bilmes, 2013); **Bilevel Optimization Based Method**: It transforms the coreset selection problem into a bilevel-optimization problem whose outer and inner objectives are subset selection and model parameter optimization, respectively: **Glister** (Killamsetty et al., 2021b). Due to the page limit, we introduce these methods and their implementation details in Appendix A and Appendix B.1, respectively.

**Implementation Details** For the LM, we follow Min et al. (2022a) to use GPT2-L (Radford et al., 2018). We set the number of retained examples of filtering $m$, the weight of diversity $\lambda$, the beam size $\mathcal{B}$ and the number of diversity search iterations as 500, 1, 8 and 10 respectively. We show the overall hyper-parameters, implementation details, analysis details and complexity analysis in Appendix B. For baselines and LENS, we run each method under 4 prompt templates over 10 random seeds (40 in total) and report the average performance with and without calibration (Zhao et al., 2021), unless otherwise specified. We show the overall templates and dataset statistics in Appendix C and D.

### 4.2 Main Results

We show the results in Table 2. We observe that our method significantly outperforms baselines on all datasets with or without calibration mechanism, which shows our method's best overall ability to find task-representative support examples across different settings and task families. Specially, our method shows better performance than the Random-Validation baseline and this directly demonstrates its non-triviality. Meanwhile, previous methods for gradient-based learning have similar performance with the Random baseline, and this

| Method | SST-2 | SST-5 | Amazon | MR | Subj | TREC | AGNews | DBPedia | Average |
|---|---|---|---|---|---|---|---|---|---|
| zero-shot | 63.0/80.3 | 27.5/33.3 | 31.2/37.6 | 61.7/77.4 | 51.0/52.0 | 38.7/27.7 | 59.8/59.9 | 32.3/37.6 | 45.6/50.7 |
| Random | 57.9/64.4 | 27.5/23.9 | 73.7/78.5 | 59.5/67.1 | 55.0/60.9 | 30.3/24.9 | 33.6/47.8 | 16.3/64.7 | 44.2/54.0 |
| Herding | 62.0/63.7 | 24.8/20.5 | 75.4/71.9 | 54.1/57.3 | 56.5/56.7 | 26.4/22.2 | 38.7/35.7 | 7.4/61.7 | 43.2/48.7 |
| K-Center Greedy | 58.6/61.6 | 25.1/23.0 | 78.6/76.3 | 59.0/61.3 | 59.9/57.0 | 31.3/26.4 | 42.3/37.8 | 32.1/72.1 | 48.4/51.9 |
| Entropy | 62.4/67.4 | 25.5/26.6 | 71.4/76.2 | 54.1/56.2 | 53.9/51.7 | 26.2/21.3 | 30.6/35.3 | 14.5/46.9 | 42.3/47.7 |
| LeastConfidence | 58.4/63.0 | 26.0/23.2 | 73.8/72.1 | 55.9/57.3 | 58.0/51.9 | 23.5/21.5 | 31.6/36.7 | 9.1.59.9 | 42.0/48.2 |
| Margin | 62.4/67.4 | 26.1/22.6 | 76.9/76.6 | 54.1/56.2 | 53.9/51.7 | 24.2/21.0 | 38.1/45.0 | 7.1/58.4 | 42.9/49.9 |
| Cal | 59.3/66.7 | 25.3/25.5 | 75.7/75.4 | 66.2/67.7 | 64.6/55.6 | 31.8/30.7 | 42.3/46.6 | 30.3/73.9 | 49.4/55.3 |
| Forgetting | 61.6/68.2 | 27.7/23.5 | 77.1/78.2 | 56.7/59.4 | 55.1/53.2 | 28.7/28.6 | 33.4/39.8 | 10.5/64.7 | 43.9/52.0 |
| GraNd | 54.6/56.5 | 27.8/24.9 | 75.5/73.4 | 52.8/55.8 | 55.7/51.9 | 28.2/23.9 | 33.4/53.2 | 21.2/64.0 | 43.7/50.5 |
| CRAIG | 63.4/72.0 | 26.4/25.3 | 75.4/80.6 | 59.3/66.7 | 57.0/54.8 | 32.0/24.8 | 37.4/55.4 | 29.5/71.0 | 47.6/56.3 |
| GradMatch | 57.0/61.9 | 26.3/23.4 | 75.1/76.0 | 56.6/62.1 | 55.8/64.6 | 25.8/21.4 | 32.6/39.4 | 9.7/57.8 | 42.3/49.6 |
| FacilityLocation | 65.5/73.2 | 23.9/24.3 | 77.7/77.6 | 61.7/69.6 | 59.0/54.5 | 35.7/29.1 | 42.5/58.6 | 30.4/74.3 | 49.6/57.7 |
| GraphCut | 65.0/71.4 | 25.3/24.6 | 76.5/78.4 | 66.3/72.3 | 63.2/56.6 | 34.7/28.4 | 41.9/50.8 | 14.0/47.3 | 48.4/53.7 |
| Glister | 59.0/61.0 | 25.8/25.2 | 78.0/77.7 | 60.1/66.6 | 60.6/57.0 | 29.4/23.4 | 38.6/41.1 | 24.1/69.8 | 46.7/52.7 |
| Random & Valid (100) | 70.1/72.8 | 33.2/29.6 | 76.3/78.4 | 53.2/57.9 | 60.7/58.9 | 30.3/24.9 | 40.1/33.6 | 37.2/56.7 | 50.1/51.6 |
| Random & Valid (1000) | 65.0/62.6 | 29.0/30.8 | 78.5/78.6 | 57.7/62.5 | 61.9/53.4 | 32.8/29.4 | 43.5/56.1 | 37.8/73.6 | 49.5/55.9 |
| LENS | **86.3/87.6** | **44.9/42.1** | **80.2/83.9** | **83.1/83.9** | **86.4/81.5** | **59.0/48.8** | **77.9/78.1** | **40.8/80.7** | **69.8/73.3** |

Table 2: Results on GPT2-L. Each entry shows the ICL accuracy without and with calibration (Zhao et al., 2021).

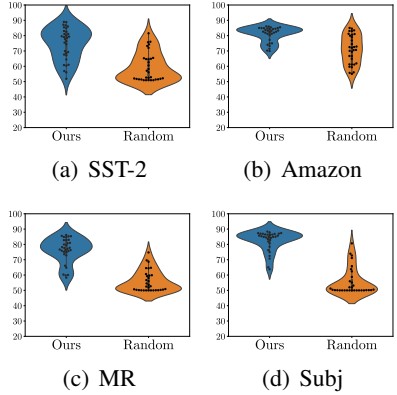

(a) SST-2      (b) Amazon

(c) MR      (d) Subj

Figure 2: The performance distribution of support examples and random examples with multiple orders.

indicates: 1. there is a non-negligible gap between ICL and these methods 2. it is necessary to design to ICL-specific method to find support examples.

Additionally, the Random baseline slightly underperforms the Zero-Shot, which shows that random examples are hard to fully characterize the task and the necessity of finding support examples for ICL. In experiments, we observe that Random-Validation suffers from the ICL's instability. Specifically, we find that considerable example permutations selected by the validation set do not consistently yield satisfactory results on the test set and this degrades its performance, whereas our method is more robust and less susceptible to this issue.

### 4.3 Analysis

**The Sensitivity of Support Examples to Orders**
The recent study (Lu et al., 2022) shows that the ordering of in-context examples for ICL has a significant influence on the performance. Specifically, to the same set of randomly sampled examples, different orders can result in near state-of-the-art and random-guess performance. In this section, we explore the effect of ordering for our support examples on SST-2, Amazon, MR and Subj. For each task, we select four sets of support examples and four sets of random examples and then evaluate their performance with eight randomly sampled orders. We show the performance distribution in Figure 2. We see that random examples with different orders show highly unstable performance where the worst drops to the random-guess level, which is consistent with the conclusion in previous work (Lu et al., 2022). In contrast, the support examples' performance is significantly more stable than random examples. Generally, most orders can still lead to approximately equivalent performance as the searched orders and few orders lead to the random-guess performance. The phenomenon is compatible with the conclusion from the recent work (Chen et al., 2022), which shows a strong negative correlation between ICL sensitivity and accuracy. Moreover, our support examples' lower sensitivity to the ordering demonstrates that they can more effectively depict and characterize the corresponding task.

**Transferablity across Different LMs** In the main experiments, we get GPT2-L's support examples and evaluate them using the same LM. And

| Examples | SST-2 | MR | TREC | AGNews | Average |
|---|---|---|---|---|---|
| *GPT2-XL* | | | | | |
| Random | 60.3 | 66.7 | 34.4 | 56.9 | 54.6 |
| LENS | 73.0 | 70.7 | 44.0 | 60.4 | 62.0 |
| *GPT2-M* | | | | | |
| Random | 59.6 | 65.0 | 33.4 | 51.7 | 52.4 |
| LENS | 82.0 | 75.5 | 37.5 | 75.3 | 67.6 |
| *GPT-Neo-2.7B* | | | | | |
| Random | 57.3 | 55.8 | 29.4 | 71.6 | 53.5 |
| LENS | 62.5 | 65.2 | 45.0 | 80.2 | 63.2 |

Table 3: The transfer of GPT2-L's support examples on LMs with different sizes and pre-training corpora: GPT2-M, GPT2-XL and GPT-Neo-2.7B.

| | SST-2 | Subj | TREC | Average |
|---|---|---|---|---|
| Random | 57.9 | 55.0 | 30.3 | 47.7 |
| $p=1.5$ | 86.6 | 86.1 | 59.5 | 77.4 |
| $p=2$ | 86.3 | 86.4 | 59.0 | 77.2 |
| $p=3$ | 85.1 | 86.0 | 58.7 | 76.6 |
| $m=50$ | 83.6 | 84.1 | 56.2 | 74.6 |
| $m=500$ | 86.3 | 86.4 | 59.0 | 77.2 |
| $m=1000$ | 86.1 | 85.9 | 59.1 | 77.0 |
| $\lambda=1$ | 86.3 | 86.4 | 59.3 | 77.3 |
| $\lambda=2$ | 86.5 | 86.2 | 58.8 | 77.2 |
| $\lambda=0.5$ | 85.9 | 86.1 | 58.7 | 76.9 |
| $\lambda=0$ | 67.5 | 68.2 | 37.7 | 57.8 |
| $\mathcal{B}=1$ | 83.1 | 76.2 | 42.2 | 67.2 |
| $\mathcal{B}=4$ | 85.9 | 86.0 | 58.5 | 76.8 |
| $\mathcal{B}=8$ | 86.3 | 86.4 | 59.0 | 77.2 |

Table 4: Impact of different hyper-parameters.

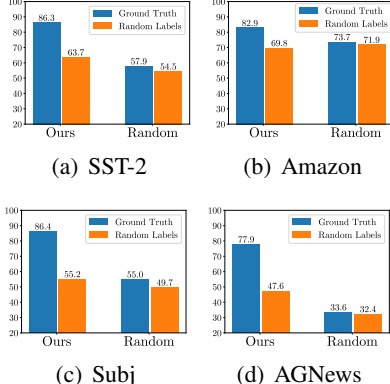

Figure 3: The results under gold and random labels.

here we explore the transferability of these support examples across different LMs with various sizes and pre-training corpora. Specifically, we test the support examples of main experiments on GPT2-M (355M), GPT2-XL (1.5B) and GPT-Neo-2.7B (Black et al., 2021). The results are shown in Table 3. We see that GPT2-L's support examples also show better performance than the Random baseline. Additionally, our support examples also demonstrate the consistent superiority on GPT-Neo-2.7B which has a different pre-training corpus from GPT2-L. Since random examples' performance can not be well transferred to other LMs (Lu et al., 2022), these can show the strong transferability of our support examples and the utility of our method when more powerful LMs are proposed.

**Ground Truth Matters for Support Examples** Recently, Min et al. (2022b) suggests that ground truth (GT) labels are not important for ICL, which differs from traditional supervised learning. In their experiments, for randomly sampled examples, using ground truth labels or not leads to similar ICL performance. Here we explore the effect of ground truth labels for support examples. We show the

performance of support examples and random examples with GT or random labels in Figure 3. We see that the results on random examples are consistent with that in the previous paper (Min et al., 2022b). However, we observe a significantly different trend in support examples. Specifically, after removing GT labels, support examples' performance gets a strong degradation. We suppose it is because while random examples can not characterize the task well and thus their GT labels are not important, the GT labels of support examples contain crucial task information and input-output correspondence, so their GT labels are important for ICL's performance. Meanwhile, we find that under random labels, support examples also yield noticeable improvements over the random examples, which indicates that the inputs of support examples are also more informative for the task.

**The Impact of Hyper-parameters** In this section, we evaluate the effect of each hyper-parameter. Specifically, we evaluate the effect of progressive ratio $p$, the number of stage 1's retained examples $m$, the weight of diversity $\lambda$ and the beam size $\mathcal{B}$ by separately tuning them and observing performance. Table 4 shows the results. When $\mathcal{B}$ is set to 0, i.e., we remove stage 2 and just select those examples with the highest InfoScore, the performance gets significantly degraded, and this directly demonstrates the effectiveness and necessity of stage 2. Except when $\mathcal{B}=0$, our method leads to consistent performance improvements compared with the Random baseline in general, across various hyper-parameter configurations, which indicates our method's robustness to hyper-parameters. Meanwhile, we observe two slight performance degradations when $m=100$ or $\mathcal{B}=1$. For

|            | Subj | AGNews |
|------------|------|--------|
| Random     | 55.0 | 33.6   |
| Filtering (Uninformative) | 52.5 | 27.4 |
| Filtering (Informative) | 65.8 | 47.8 |
| Filtering + Search | 86.4 | 77.9 |

Table 5: The Effect of Progressive Filtering.

the case that $m = 100$, we suppose that is because there are too few examples being retained after stage 1, limiting candidate examples' diversity. When $\mathcal{B} = 1$, our stage 2 degrades to greedy search guided by the diversity, causing it susceptible to the local optimum issue.

**InfoScore and Progressive Filtering**  In this section, we evaluate the effect of InfoScore and progressive filtering in stage 1. Specifically, we randomly sample examples from the retained examples of stage 1 and test their average performance across 4 prompt templates with 10 random orders (40 in total). We compare the Random baseline, our filtering method and another filtering variant that filters uninformative examples, which retains those examples with *low* InfoScore at each iteration. We show the results in Table 5. We observe that just the proposed filtering method also leads to better ICL performance than randomly sampled examples, which directly shows that stage 1 is effective for filtering out the uninformative examples. Meanwhile, the performance points of Filtering (Informative), Random and Filtering (Uninformative) present a descending trend, which demonstrates that the proposed InfoScore can indicate the examples' in-context informativeness. However, compared with our entire method, Filtering (Informative) still shows a significant discrepancy. This indicates the necessity of considering in-context examples' dependency and the effectiveness of the proposed diversity-guided search.

## 5   Related Work

Since we introduce a wide range of coreset selection methods in Section 4.1, we omit them here and mainly introduce previous works about example selection for ICL. Previous works mainly consider example-level retrieval for ICL. Liu et al. (2022) leverage a semantic embedder to retrieve relevant examples for the given test input. Das et al. (2021) and Hu et al. (2022) use dense retrievers trained by task-specific targets' similarities to retrieve in-context examples for question answering and dialogue state tracking, respectively. Rubin et al. (2022); Shi et al. (2022) train the demonstration retriever based on the feedback of the language model for semantic parsing. Wu et al. (2022) use Sentence-BERT (Reimers and Gurevych, 2019) to retrieve relevant examples and introduce an information-theoretic-driven criterion to rerank their permutations. Levy et al. (2022); Ye et al. (2023) further consider diversity in example retrieval. Different from these methods which aim to provide example-specific information for the test input, we focus on task-level example selection, which seeks to find examples that are representative for the task and is complementary to them. Moreover, because the large language models (Brown et al., 2020; Zhang et al., 2022a; Black et al., 2021) almost adopt purely causal Transformer (Vaswani et al., 2017) decoder architecture, we can calculate task-level in-context examples representation in advance and reuse them for different test inputs. Since these two settings' goals are orthogonal and complementary, we regard the hybrid setting and method as future work. Another line of methods is active learning (Ren et al., 2022) for ICL. It aims to select some examples from a large pool of unlabeled data and annotate them for ICL. Zhang et al. (2022b) propose to learn an active example selector by off-line reinforcement learning and use it to select examples to annotate for ICL. Su et al. (2022) propose a graph-based annotation method, vote-k, and use Sentence-BERT to retrieve relevant examples from the annotated examples for ICL. In this paper, we explore a different setting for ICL's example selection, where we select support examples from the annotated dataset since there are massive annotated datasets for various tasks and the prevailing large language model has shown impressive data annotation ability (Efrat and Levy, 2020; Gao et al., 2022; Ye et al., 2022; Meng et al., 2022; Cheng et al., 2023).

## 6   Conclusion

In this paper, we propose a two-stage filter-then-search method to find support examples for in-context learning from the annotated dataset: First we propose InfoScore to select informative examples individually with a progressive filtering process. Then we propose diversity-guided example search which iteratively refines and evaluates the selected examples, to find the example permutations that fully depict the task. The experimental results show that our method significantly outperforms extensive baselines, and further analyses show that each component contributes critically to

the improvements and shed light on the principles of support examples and in-context learning.

## Limitations

These are the limitations of this work:

- Due to the computation resources limitation, we mainly conduct experiments on GPT2-L (Radford et al., 2018) and analyze the cross-LM transferability of support examples in section 4.3. We see the exploration on more LMs as future work.

- In this paper, the proposed filter-then-search framework explores how to find support examples of in-context learning. We see exploring and analyzing more principles of in-context learning as future work.

- Language models have exhibited various kinds of bias (Bender et al., 2021), since our filtering stage is based on its feedback, the filtered example might also exhibit these biases. We see language model debiasing as an important future research topic.

## Acknowledgements

This work was supported by the National Natural Science Foundation of China (No. 62236004 and No. 62022027).

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

## A Baselines

We mainly compare our proposed methods with following baselines: **Random**: We randomly select examples with random orderings from the training set; **Random & Validation**: We evaluate multiple sets of random examples on the validation set and select the best one. We consider Random & Validation under two settings whose computational cost is similar to our method: 1. the size of validation set is the same as ours at stage 2 (100) and the number of random example sets is the same as our searched and evaluated example permutations (640). 2. the size of validation set is larger, 1000, and the number of random example sets is 100. We also consider a wide range of **Coreset Selection** methods in traditional gradient-based learning scenarios, according to the methodologies, they can be divided into multiple categories including: **Geometry-Based Method**: it assumes that data points that are close in the feature space have similar properties. The **Herding** method (Chen et al., 2012) adds one data point each time into the coreset to greedily minimize the distance between coreset center and the original dataset center. The **K-Center Greedy** method (Sener and Savarese, 2018) selects examples to minimize the largest distance between each example in coreset and its closest example in set of examples that are not in coreset. **Uncertainty-Based Method**: it assumes examples with higher uncertainty can have a greater impact on the model and should be contained in coreset. Coleman et al. (2020) propose **Least Confidence** (the max probability over all labels), **Entropy** and **Margin** (max probability margin between different labels) to measure the examples' uncertainty scores and build coreset. **CAL** (Margatina et al., 2021) selects examples whose predictive likelihood exhibits the greatest divergence from their neighbors to build the coreset. **Error/Loss Based Method**: It assumes the example that contributes more to the error or loss during training is more important and should be added to coreset. Toneva et al. (2019) propose **Forgetting** to evaluate each example's importance by counting how many times it is forgotten, i.e., it is misclassified after begin correctly classified in previous training epochs. Paul et al. (2021) propose the **GraNd** score to select informative examples. GraNd is the gradient norm expectation of the example. The larger one example's GraNd is, the more important it is. **Gradient Matching Based Method**: Since deep models are usually trained by gradient descent, it tries to find a coreset whose gradients can imitate the entire dataset's gradients. Mirzasoleiman et al. (2020) propose **CRAIG** to convert the gradient matching problem to the maximization of a monotone submodular function and optimize it greedily. Killamsetty et al. (2021a) propose **GradMatch** based on CRAIG, which adds a regularization term to discourage assigning large weights to individual examples and improves the used greedy algorithm. **Submodularity-Based Method**: Submodular functions (Iyer and Bilmes, 2013) naturally measure the subset's informativeness and diversity and thus can be powerful for coreset selection. Iyer and Bilmes (2013) leverage **Facility Location** and **Graph Cut** as submodular functions to select the coreset. **Bilevel Optimization Based Method**: It transforms the coreset selection problem into a bilevel-optimization problem whose outer and inner objectives are subset selection and model parameter optimization, respectively. **Glister** (Killamsetty et al., 2021b) leverages a validation set on the outer optimization and the log-likelihood in the bilevel optimization.

To reduce the gap between these methods and ICL, we use the same LM (GPT2-L) with "last pooling" fine-tuned on the whole dataset for 5 epochs to obtain relevant metrics, e.g., gradients or forgetting times for these methods. Following Guo et al. (2022), we use the gradients of the final fully-connected layer's parameters as these methods' example feature.

## B Implementation Details

### B.1 Baseline Details

For those previous coreset selection methods, to reduce the gap between these methods and ICL, we use the same LM (GPT2-L) with "last pooling" fine-tuned on the whole dataset for 5 epochs to obtain relevant metrics, e.g., gradients or forgetting times for these methods. Following Guo et al. (2022), we use the gradients of the final fully-connected layer's parameters as these methods' example feature. For baselines that output a weighted subset of examples, e.g., CRAIG or GradMatch, we just adopt its examples for simplicity since there are few methods to weighting in-context examples for ICL.

## B.2 Method Details

We find each prompting template's corresponding support examples separately in our method and compared methods, i.e., we select examples for each prompting template separately. For simplicity, we calculate Eq (3) without calibration for experiments without or with calibration. For the filtering stage, we set the progressive factor and the size of initial score set according to the dataset's size. Specifically, we set the progressive factor to make the filter iterations be 4.

We run all experiments under the label balance setting and the total number of in-context examples for most datasets except DBPedia is set to 8. The number of some datasets' in-context examples is not 8 but close to 8 because 8 can not be divided by the number of its labels, e.g., 5 for SST-5. Since DBPedia has 14 labels and significantly longer input sequence, we run experiments on it under the label-unbalance setting and set the total number of examples to 4. In label balance setting, we 1. filter the same number of examples for each label, 2. initialize the example permutation of stage 2 with balanced labels 3. update $e^*$ with $e^*_{new}$, whose label is the same as $e^*$.

We list the total number of examples in Table 6. And we set the size of initial score set to make the times of LM's forwards to be around 10K. We list the progressive factor $p$ and the size of initial score set $|S_0|$ in Table 7. For other hyper-parameters, we conduct grid search for the number of retained examples of filtering $m$, the weight of diversity $\lambda$, the beam size $\mathcal{B}$ and the iteration of diversity-guided search over {500,1000}, {0.5,1,2}, {4,8,16} and {5,10,15} respectively on the SST-2 dataset. And we set them to be 500, 1, 8 and 10 respectively.

## B.3 Experimental Details

In section "The Sensitivity of Support Examples to Orders", since the performance is sensitive to the prompting templates, we show the performance distribution under a specific prompting template. In other analysis experiments, for simplicity, we report the average performance under four different prompting templates without calibration, unless otherwise specified.

### B.3.1 The Complexity of Our Method

**Progressive Filtering**  in the filtering stage, we need to compute pairwise Eq 3 for $N*l*\rho/\rho = N*l$ times, where $N$ is size of training set. Since we filter the dataset into $1/\rho$ of its previous size until a

|  | Training Size | Test Size | Label | Examples |
|---|---|---|---|---|
| SST-2 | 6921 | 873 | 2 | 8 |
| SST-5 | 8544 | 2210 | 5 | 10 |
| Amazon | 30000 | 2000 | 2 | 8 |
| MR | 8662 | 2000 | 2 | 8 |
| Subj | 8000 | 2000 | 2 | 8 |
| TREC | 5452 | 500 | 6 | 12 |
| AGNews | 30000 | 7601 | 4 | 8 |
| DBPedia | 30000 | 2000 | 14 | 4 |

Table 6: Data statistics.

small set of examples is left, the number of iteration is $log_{\rho}N$). Thus the filtering stage's complexity over $N$ is $O(N * \log_{\rho} N)$. In experiments, we set $\rho$ to $N^{\frac{1}{C}}$ to make it a linear complexity, where $C$ is a constant. According to the size of dataset, $\rho$ is usually set between 2 - 3, shown in Table 7.

**Diversity-Guided Example Search**  At each iteration, we have $\mathcal{B}$ candidate permutations and separately update them $B$ times. And then we evaluate these updated candidate permutations on the small validation set sampled from the remaining training set, whose size is fixed. Since updating the candidate permutations reuses the intermediate results of the filtering stage and does not involve the computation of the LLM (see Eq (4) and (5)), we omit it for complexity analysis. So the complexity of diversity-guided example search is consistant, $\mathcal{B} * \mathcal{B}$.

## C Prompting Templates

We show the prompting verbalizers and templates in Table 8.

## D Dataset Split and Statistics

We use the same dataset split in the previous work (Min et al., 2022a). Due to computational resource limitations, for Amazon, AGNews and DBPedia, we conduct experiments on a randomly sampled subset of it (30000 and 2000 for the training and test set), and we show the overall dataset statistics in Table 6.

|         | $p$ | $|S_0|$ |
|---------|-----|---------|
| SST-2   | 2   | 14      |
| SST-5   | 2   | 11      |
| Amazon  | 3   | 4       |
| MR      | 2   | 11      |
| Subj    | 2   | 12      |
| TREC    | 2   | 20      |
| AGNews  | 3   | 4       |
| DBPedia | 3   | 4       |

Table 7: The progressive factor $p$ and the size of initial score set $S_0$ for each dataset.

---

**Task Family: Sentiment Classification**

---

**Task**: SST-2
**Prompting Verbalizer**: {great, terrible}
**Prompting Templates**:

- "[INPUT] A [VERBALIZER] one. "

- "[INPUT] It was [VERBALIZER]. "

- "[INPUT] All in all [VERBALIZER]. "

- "[INPUT] A [VERBALIZER] piece. "

Example:
**Input:**
I have to admit that I am baffled by jason x.
*It was terrible.*
If you answered yes, by all means enjoy the new guy.
*It was great.*
. . .
Never comes together as a coherent whole.
*It was*
**Output:**
*terrible.*

---

**Task**: SST-5
**Prompting Verbalizer**: {great, good, okay, bad, terrible}
**Prompting Templates**: Same as SST-2

---

**Task**: Amazon
**Prompting Verbalizer**: {great, good, okay, bad, terrible}
**Prompting Templates**: Same as SST-2

---

**Task**: MR
**Prompting Verbalizer**: {great, terrible}
**Prompting Templates**: Same as SST-2

---

**Task Family: Topic Classification**

---

**Task**: TREC
**Prompting Verbalizer**: {Description, Entity, Expression, Human, Location, Number}
**Prompting Templates**:

- "[INPUT] Topic: [VERBALIZER]. "

- "[INPUT] Subject: [VERBALIZER]. "

- "[INPUT] This is about [VERBALIZER]. "

- "[INPUT] It is about [VERBALIZER] piece. "

Example:
**Input:**
How do storms form ?
*Topic: Description.*
What city in Florida is Sea World in?
*Topic: Location.*
. . .
What university fired Angela Davis?
*Topic:*
**Output:**
*Human.*

---

**Task**: AGNews

**Prompting Verbalizer**: {World, Sports, Business, Technology}
**Prompting Templates**: Same as TREC

---

**Task**: DBPedia
**Prompting Verbalizer**: {Company, Educational Institution, Artist, Athlete, Office Holder, Mean of Transportation, Building, Natural Place, Village, Animal, Plant, Album, Film, Written Work}
**Prompting Templates**: Same as AGNews

---

**Task Family: Subjective Classification**

---

**Task**: Subj
**Prompting Verbalizer**: {subjective, objective}
**Prompting Templates**:

- "[INPUT] This is [VERBALIZER]. "

- "[INPUT] It's all [VERBALIZER]. "

- "[INPUT] It's [VERBALIZER]. "

- "[INPUT] Is it [VERBALIZER]? "

Example:
**Input:**
There are two distince paths in life good vs . evil. *It's subjective.*
Photographed with melancholy richness and eloquently performed yet also decidedly uncinematic.
*It's objective.*
· · ·
The film is an homage to power , strength and individualism.
*It's*
**Output:**
*subjective*

Table 8: The prompting verbalizers and templates for each task.