# OpenReview forum: "Finding Support Examples for In-Context Learning"
_EMNLP/2023/Conference — EMNLP 2023 Findings_

### Official Review · Reviewer_Hn2j · 2023-07-23

**Soundness:** 3

**Excitement:**

4: Strong: This paper deepens the understanding of some phenomenon or lowers the barriers to an existing research direction.

**Missing References:**

[1] GPS: Genetic Prompt Search for Efficient Few-shot Learning, Shao et al, EMNLP'22 - for constructing task specific prompts
[2] Fantastically Ordered Prompts and Where to Find Them: Overcoming Few-Shot Prompt Order Sensitivity, Lu et al, ACL'22 - ordering of samples
[3] Reordering Examples Helps during Priming-based Few-Shot Learning, Kumar et al, ACL findings'21


**Paper Topic And Main Contributions:**

The paper addresses the problem of selecting the best prompt for in-context learning with a large labeled dataset. In this paper, they focus on finding the best task specific prompt, independent of the inference sample, without having access to the gradients. The solution is based on a filter-then-search pipeline, which first extracts the most "informative" samples from the training set. Then, it reorders the top informative samples to construct the prompt. The paper provides an extensive experimental study on 8 datasets on more classical approaches showing state-of-the-art results on task specific prompt selection. In addition, the paper shows that the selected examples are more robust to ordering and changing the model than the random approach.

**Questions For The Authors:**

A. In algorithm 1, you define 'l'. Which 'l' did you use in the experiments?
B. Algorithm 1, line 11, did you mean the top |D'|/ρ samples? The current line leads to selecting the top ~1/N^0.5 samples which doesn't make sense.

**Reasons To Accept:**

1. Simple yet effective solution which outperforms the other task-specific baselines in the paper - the task-specific setup will fit an online system with very low latency inference (to avoid heavy computation for inference sample oriented solutions). Moreover, the linear algorithm to calculate the InfoScore can be beneficial to focusing on informative samples.
2. Extensive experimental study showing state of the art results in addition to ablation test showing less sensitive to sample order
3. Paper is easy to follow

**Reasons To Reject:**

1. The baselines are classical while there is missing state-of-the-art related work[1,2,3] published last year which can act as better baselines
2. Unclear how well it generalizes to IFT models - the solution uses GPT2 LMs. However, more advanced solutions such as Instruction Fine-Tuned (IFT) like Flan-T5 are not being evaluated. Thus, it is unclear how well this solution keeps up with the existing IFT solutions.
3. Missing runtime comparison - the solution seems to be more exhaustive than other baselines. The runtime cost of the algorithm was not reported as part of the paper.

[1] GPS: Genetic Prompt Search for Efficient Few-shot Learning, Shao et al, EMNLP'22 - for constructing task specific prompts
[2] Fantastically Ordered Prompts and Where to Find Them: Overcoming Few-Shot Prompt Order Sensitivity, Lu et al, ACL'22 - ordering of samples
[3] Reordering Examples Helps during Priming-based Few-Shot Learning, Kumar et al, ACL findings'21

**Reproducibility:**

4: Could mostly reproduce the results, but there may be some variation because of sample variance or minor variations in their interpretation of the protocol or method.

**Reviewer Confidence:**

3: Pretty sure, but there's a chance I missed something. Although I have a good feel for this area in general, I did not carefully check the paper's details, e.g., the math, experimental design, or novelty.

**Typos Grammar Style And Presentation Improvements:**

- The related work should be moved to the beginning. In the current format it was hard to follow on the positioning compared to other related work which only made clearer after reading the RW section.
- In line 533 you mention m=100 and refer to table 4, which doesn't include the m=100 setting.

---

> ### Author Rebuttal · Authors · 2023-08-29
>
> Thank you for your comprehensive review and insightful suggestions. Below is our response to the comments and questions.
>
> **Q1**: Will we add baselines and experiments on newer language models?
>
> Yes, we will add the state-of-the-art related work as baselines and add the experiments on the latest language model (LLaMA2, LLaMA2-chat) in the next version.
>
> **Q2**: The runtime comparison.
>
> The runtime of our method is comparable with baselines. Our proposed method only depends on the inference of the language model. But baselines depend on finetuning the language model, which requires more strong computational resources and more time. We will add the detailed runtime report in the next version. Meanwhile, the examples selected by our method lead to significant improvements over baselines.
>
> **Q3**: Question for symbols in Algorithm 1:
>
> About the "l":
>
> We list the value of "l" for each dataset in Appendix B.2 (Table 7).  We will adjust the position of Table 7 and improve the layout to make it more clear.
>
> The line 11 in Algorithm 1:
> Yes, we meant the top |D'|/ρ samples. We will fix the presentation in the next version.
>
> **Q4**: typos and presentation improvements
>
> 1) The position of the section "Related Work"
>
> Thank you for your detailed review and pointing out these issues. We will move the section of related work to the beginning (after introduction).
>
> 2) The typo in Table 4:
>
> The entry 'm=500' in Table 4 is a mistake, which should be "m=100", we will fix it in the next version.
>
> **Q5**: Missing References
>
> We will include citations for the papers you have suggested in the next version.
>
> Thank you again for your valuable review. If we have misunderstood any aspect of your review, or if you have additional questions, please let us know.

---

### Official Review · Reviewer_wMx9 · 2023-08-04

**Soundness:** 3

**Excitement:**

3: Ambivalent: It has merits (e.g., it reports state-of-the-art results, the idea is nice), but there are key weaknesses (e.g., it describes incremental work), and it can significantly benefit from another round of revision. However, I won't object to accepting it if my co-reviewers champion it.

**Missing References:**

[1] Understanding Black-box Predictions via Influence Functions, ICML 2017

[2] Estimating Training Data Influence by Tracing Gradient Descent, NIPS 2020

[3] Tracing Knowledge in Language Models Back to the Training Data, Findings of EMNLP2022

[4] Understanding In-Context Learning via Supportive Pretraining Data, ACL2023


**Paper Topic And Main Contributions:**

This paper proposes a new topic that finding support examples for in-context learning, which could well characterize the task for in-context learning. To select such support examples, this paper proposes a two-stage fiLter-thEN-Search (LENS) method. In the first stage, this paper proposes InfoScore metric, which could help filter the dataset to obtain informative in-context examples. In the second stage, this paper proposes a diversity-guided search method that iteratively refines and selected examples to find support examples. Extensive experiments show that LENS achieve better performance that traditional coreset selection methods.

**Questions For The Authors:**

A: All experiments are conducted on the fixed shot. The number of shots indeed has a significant impact on ICL performance, and there are several methods that perform well only on certain shots. Would LENS also face such issues? Testing under different shots would provide more convincing results.

B: What is the label distribution of support in-context examples? Balance or not balance?

C: Figure 3 shows the results under gold and random labels. This experiment is motivated by Min et.al (2022b). Actually, Yoo et.al (2022) (Ground-Truth Labels Matter: A Deeper Look into Input-Label Demonstrations, EMNLP2022) shows that golden labels matter in some tasks. Are the test tasks in Figure 3 originally sensitive to the golden labels? Maybe comparison with other examples could answer this question.

D: In Section 3.1, this paper test InfoScore on the score set. What the impact of this score set? Intuitively, the similarity between the score set and the test set is crucial, higher similarity could lead to higher test performance. Do the actual results perform so? This paper randomly sample score set (Line 2 in Algorithm 1), what is the standard deviation?


**Reasons To Accept:**

- Finding in-context examples that could well characterize the task is an important topic and would benefit the future research. And this paper focus on this topic.
- The paper is well-organized. The proposed two-stage method LENS is described clearly. In the first stage, the defined InfoScore evaluates the informativeness of an example and could help filter the original dataset to obtain informative in-context examples. The second stage further introduces diversity into the selected informative examples, which is also well-motivated. The experiments also show the effectiveness of LENS.
- The further analyses show the extra characteristic of support in-context examples: less sensitivity to order, ground truth labels matter, and good transferability across LMs.


**Reasons To Reject:**

- The experiments are all conducted on GPT2-L, which is not enough. More experiments on other LMs are required. Although the authors claims that this is because of the computation resources limitation. Actually, all experiments in this paper only require inference, other small LMs such as GPT2-xl, which could be inferenced on a single 2080Ti/3090Ti. In summary, experiments only on GPT2-L lack persuasiveness, more experiments on other LMs (at least extra one) would improve this paper.
- The “support in-context examples” seems to be misleading, especially the “support”. The similar idea has been widely studied in interpretability, especially the instance-based explanations ([1][2]). More recently, similar topic is also studied in prompt learning [3] and in-context learning [4]. The coreset selection methods also make connections between training examples and test set and this paper mainly compare with these methods. Actually, this paper mainly focuses on select examples which could characterize the task. In essence, this paper focuses on select examples which could stand for the training set. The only aspect that demonstrates supportiveness is the InfoScore. Therefore, the “support” may be not confusing and not accurate.
- Although this paper provides some analyses on support in-context examples, it is not enough. It would be more meaningful to examine the characteristics of these in-context examples, such as label distribution, word distribution [4], and so on. These factors can greatly assist in better understanding the in-context learning.


**Reproducibility:**

4: Could mostly reproduce the results, but there may be some variation because of sample variance or minor variations in their interpretation of the protocol or method.

**Reviewer Confidence:**

4: Quite sure. I tried to check the important points carefully. It's unlikely, though conceivable, that I missed something that should affect my ratings.

---

> ### Author Rebuttal · Authors · 2023-08-29
>
> Thank you for your comprehensive review and insightful suggestions. Below is our response to the comments and questions.
>
> **Q1**: Experiments on LLaMA models
>
> We agree with you that it is important to verify the method on the latest strong language models and we will add the corresponding experiments (LLaMA2, LLaMA2-chat) in the next version.
>
> **Q2**: About the name of the proposed method
>
> Thank you for your detailed opinions on our method name. We will consider renaming in the next version.
>
> **Q3**: More analyses of the selected examples
>
> We conduct analytic experiments on the sensitivity of support examples to orders, transferability of support examples across different LMs, the effect of ground truth to support examples, hyper-parameter sensitivities and the ablation study about the proposed components. We plan to extend these analyses to cover word distribution and the gradients of support examples in the next version.
>
> **Q4**: The performance of our method under different demonstration quantities
>
> In the primary experiments, we have standardized the demonstration quantity to ensure consistent inference costs across all evaluated methods. Here are our method's results (without calibration) on three datasets, SST-2, Subj, and Amazon, using varying numbers of demonstrations:
>
> **SST-2**
>
> Demonstration Quantity (per Label) | 1    | 2    | 3    | 4    |
> |----------------------------------------|------|------|------|------|
> Accuracy                               | 85.2 | 86.9 | 86.1 | 86.3 |
>
> **Subj**
>
> Demonstration Quantity (per Label) | 1    | 2    | 3    | 4    |
> |----------------------------------------|------|------|------|------|
> Accuracy                               | 82.0 | 85.9 | 86.2 | 86.4 |
>
> **Amazon**
> Demonstration Quantity (per Label) | 1    | 2    | 3    | 4    |
> |----------------------------------------|------|------|------|------|
> Accuracy                               | 78.1 | 78.7 | 79.3 | 80.2 |
>
> We see that our method's performance degrades slightly with fewer demonstrations, which shows the stability of our method across different demonstration quantities and superior performance over baseline methods.
>
>
>
>
>
>
> **Q5**: Are support examples label-balanced?
>
> As mentioned in the section of Implementation details (Appendix B.2), we keep using the label balance setting in all the experiments except DBPedia (The number of some datasets’ in-context examples is not 8 but close to 8 because 8 can not be divided by the number of its labels, e.g., 5 for SST-5. Since DBPedia has 14 labels and significantly longer input sequence, we run experiments on it under the label-unbalance setting and set the total number 1096 of examples to 4.).
>
> Our preliminary investigations reveal two main findings: 1) For datasets with imbalanced labels, like TREC, a balanced selection of examples significantly boosts performance. 2) For relatively label-balanced datasets, the performance remains stable irrespective of label balance. We will add these results in the next version and further analyze the effect of balanced labels.
>
> **Q6**: The comparison in Figure 3 (Ground Truth Matters for Support Examples). Are the test tasks in Figure 3 originally sensitive to the golden labels?
>
> We agree with you that it is important to compare with the original results and we have the comparison between support examples and randomly selected examples in Figure 3. Compared with random examples, we observe a significantly different trend in support examples. Specifically, after removing GT labels, support examples’ performance significantly degrades, while the performance of random examples remains similar to their original points. We will make the presentation more clear in the next version.
>
> **Q7**: 1. Does the score set influence the final performance? (Does the score set that is similar to the test set lead to better performance?) 2. The random variance of our method
>
> 1. Yes, but we cannot get access to information of the test set before the test stage. So we cannot improve the performance according to this.
>
> 2. As mentioned in line 407~409, we run our experiments under 4 prompt templates over 10 random seeds (40 in total) and report the average performance with and without calibration (Zhao et al., 2021). For all datasets, the performance of our method fluctuates within three points under the same prompt template, which is significantly more stable than the randomly selected examples and baselines. We will add the detailed standard deviation in the next version.
>
>
> Thank you again for your valuable review. If we have misunderstood any aspect of your review, or if you have additional questions, please let us know.

---

### Official Review · Reviewer_NXgr · 2023-08-12

**Soundness:** 3

**Excitement:**

3: Ambivalent: It has merits (e.g., it reports state-of-the-art results, the idea is nice), but there are key weaknesses (e.g., it describes incremental work), and it can significantly benefit from another round of revision. However, I won't object to accepting it if my co-reviewers champion it.

**Paper Topic And Main Contributions:**

This work tried to address the challenge of selecting support examples that can improve the performance of in-context learning. In-context learning is a new paradigm where a language model observes a few examples and directly predicts the test input, but it is sensitive to the provided examples. The authors propose a two-stage method, LENS (fiLter-thEN-Search), to tackle this challenge. First, the dataset is filtered to obtain informative in-context examples individually using a novel metric called InfoScore. Then, a diversity-guided example search is used to iteratively refine and evaluate selected example permutations. Experimental results show that LENS significantly outperforms various baselines, with further analyses highlighting the importance of each component in the method and providing insights into the principles of support examples and in-context learning.

**Questions For The Authors:**

1. Can LENS outperform dynamic selection methods (such as [1])?

2. Can LENS help larger models such as GPT-3?

**Reasons To Accept:**

1. Novel Approach: This work provides a novel approach (LENS) towards improving the performance of in-context learning.
(LENS) approach to tackle the challenge of finding support examples in two stages, by filtering the dataset and then using diversity-guided example search.

2. Extensive experimental results: The paper provides a comprehensive set of experimental results showing that LENS significantly outperforms a wide range of baselines, demonstrating the effectiveness of the proposed method.

**Reasons To Reject:**

The main limitation of this work lies in the nature of static example selection, as it can hugely limit the upper-bound performance of this method.

For selecting better in-context examples, one standard method is to dynamically select similar examples for different test cases [1].
Such a dynamic selection can make better use of the large-scale example bank.
In this work, the proposed LENS just chooses one set of prompting examples for each task.
It might not be reasonable that all cases in one task consistently prefer the same examples, especially for more complex tasks (such as reasoning tasks) in which the cases have significant diversity.

Moreover, the in-context learning is the emergent capability on models with >10B size.
However, this work mainly experiments with relatively small models, which limit the experimental contribution of this work.

[1] J. Liu, D. Shen, Y. Zhang, B. Dolan, L. Carin, and W. Chen. What makes good in-context examples for GPT-3?

**Reproducibility:**

4: Could mostly reproduce the results, but there may be some variation because of sample variance or minor variations in their interpretation of the protocol or method.

**Reviewer Confidence:**

5: Positive that my evaluation is correct. I read the paper very carefully and I am very familiar with related work.

---

> ### Author Rebuttal · Authors · 2023-08-29
>
> Thank you for your comprehensive review and insightful suggestions. Below is our response to the comments and questions.
>
> **Q1**: Can static example selection outperform dynamic example selection?
>
> We believe that static and dynamic example selection (or example retrieval) serve complementary rather than mutually exclusive roles. Our static example selection is designed to choose instances that are broadly representative of the task at hand, whereas example retrieval aims to provide example-specific information. We consider the hybrid setting as future work.
>
> Additionally, static example selection offers greater deployment efficiency and lower latency compared to dynamic example retrieval. As noted on lines 593-595 of our paper, static selection allows for the pre-computation of task-level, in-context example representations that can be reused across various test inputs.
>  In contrast, dynamic retrieval methods require an additional retrieval step, often relying on dense encoders, and must compute in-context example representations on-the-fly. Therefore, our static approach demands lower computational costs and offers quicker response times.
>
>
> From recent work on  example retrieval, we find our proposed static example selection shows competitive performance with example retrieval methods. Take the recent paper "Self-adaptive In-context Learning (ACL 2023)"**[1]** as an example (which dynamically retrieves relevant examples and adjusts their permutation according to the test input), the following is the comparison between our method and theirs:
>
> |               | Language Model | SST-2 | SST-5 | Trec  | AgNews | Avg   |
> |---------------|:----------------:|:-------:|:-------:|:-------:|:--------:|:-------:|
> | Self-Adaptive | GPT2-XL        | 91.5  | 40.3 | 42.5 | 87.9  | 65.6 |
> | Ours          | GPT2-L         | 87.6  | 44.9  | 59.0  | 78.1   | **67.4**  |
>
> We see our method (on GPT2-L 750M) outperforms Self-Adaptive example selection (On GPT2-XL 1.5B) on average and is comparable with it in general, which is the latest competitive method of dynamic example selection.
>
> **Q2**: 1) Discussion about the emergence of ICL in GPT-2; 2) Except GPT2-L, can the proposed method help larger models?
>
> We agree with you that it is important to test our method on the models > 10B.  We will add the experiments on large language models like Flan-T5, Text-davinci-003 and GPT-3.5-turbo in the next version.
>
> Our choice of GPT2-L as the inference model aligns with prior studies **[2][3][4]**, which have demonstrated that language models of the GPT-2-L scale have emerged the ability of in-context learning. And our experimental results also show that our selected example significantly outperforms randomly selected examples.
>
> In section 4.3 "Transferability across Different LMs" (line 471), we show that the examples selected by our method on GPT2-L can lead to improvements on GPT2-XL and GPT-Neo-2.7B. These can show the strong transferability of our support examples and the utility of our method when more powerful LMs are proposed.
>
>
>
> **[1]** Self-adaptive In-context Learning
>
> **[2]** MetaICL: Learning to Learn In Context
>
> **[3]** Noisy Channel Language Model Prompting for Few-Shot Text Classification
>
> **[4]** Rethinking the Role of Demonstrations: What Makes In-Context Learning Work?
>
> Thank you again for your valuable review. If we have misunderstood any aspect of your review, or if you have additional questions, please let us know.

---

### Meta-Review · Area_Chair_tTUh · 2023-09-28

**Recommendation:** 3

**Metareview:**

The authors propose a method, LENS (fiLter-thEN-Search), for selecting and ordering 'support' examples used as part of an in-context learning (ICL) prompt in LLMs in an effort to maximize performance. Specifically, LENS, first filters the dataset to obtain informative in-context examples using a newly developed InfoScore metric followed by a diversity-guided example search is used to iteratively refine selected 'support' example permutations. Experimental results on eight widely used text classification datasets show that LENS significantly outperforms several ICL example selection baselines, with additional experiments addressing questions regarding sensitivity to example ordering, selection generalization across LLMs, the value of ground truth quality in support examples, and algorithm hyperparameter sensitivity.

== Quality ==
Overall, the reviewers believe that this submission is of solid quality. Specifically, the method is conceptually well motivated and there are extensive experimental results in terms of baseline methods and additional experiments to better understand the dynamics of the proposed method. However, there were some recommendations regarding stronger experiments including larger foundational models (including instruction-tuned) additional recent applicable baseline methods, and tasks beyond text classification. Overall, the experiments support the stated premises and result in interesting findings to motivate further work.

== Clarity == Overall, the paper is well-organized and easy to understand with the appendices adding clarity. All of the reviewers were confident that they would be able to reproduce the empirical results and though the evidence as presented was convincing. The only conceptual clarification was the use of the term "support examples" (which are specific to learning discriminative functions) vs. "core set" (which align more closely with representing example distributions) -- a technical distinction, but worth considering the precision of the methodological claims. Also, while not a concern of the reviewers, discussion of theoretical implications and contextualization of the algorithmic approximations would increase overall understanding.

== Originality == While there has been work in selecting examples for ICL, the reviewers believed the specific approach to be novel and pragmatic (especially when considering algorithmic simplifications). This is a new area and there is the right balance of good conceptual motivation and heuristics to create a feasible implementation.

== Significance == As stated above, this is a increasingly widely-studied area (example selection in ICL) where small improvements can have large impact. Thus, there is definitely potential for impact, minimally as a baseline for future work. That being said, there were several concerns that the reviewers thought may hinder wide adoption including only using GPT-2 scale models (thus, without instruction-tuning and knowing that these results scale to SotA settings), only performing static selection (which is a natural area for future research with a multi-stage method), and questions regarding computational resources needed to perform this in practice. Thus, while the reviewers concur that the method is likely useful, it isn't clear what the impact will be with stronger LLMs (and if this ends up being a negative result, the impact will clearly be limited).

---

### Decision · Program_Chairs · 2023-10-07

**Decision:**

Accept-Findings

**Comment:**

The authors propose a method, LENS (fiLter-thEN-Search), for selecting and ordering 'support' examples used as part of an in-context learning (ICL) prompt in LLMs in an effort to maximize performance. Specifically, LENS, first filters the dataset to obtain informative in-context examples using a newly developed InfoScore metric followed by a diversity-guided example search is used to iteratively refine selected 'support' example permutations. Experimental results on eight widely used text classification datasets show that LENS significantly outperforms several ICL example selection baselines, with additional experiments addressing questions regarding sensitivity to example ordering, selection generalization across LLMs, the value of ground truth quality in support examples, and algorithm hyperparameter sensitivity.

== Quality ==
Overall, the reviewers believe that this submission is of solid quality. Specifically, the method is conceptually well motivated and there are extensive experimental results in terms of baseline methods and additional experiments to better understand the dynamics of the proposed method. However, there were some recommendations regarding stronger experiments including larger foundational models (including instruction-tuned) additional recent applicable baseline methods, and tasks beyond text classification. Overall, the experiments support the stated premises and result in interesting findings to motivate further work.

== Clarity == Overall, the paper is well-organized and easy to understand with the appendices adding clarity. All of the reviewers were confident that they would be able to reproduce the empirical results and though the evidence as presented was convincing. The only conceptual clarification was the use of the term "support examples" (which are specific to learning discriminative functions) vs. "core set" (which align more closely with representing example distributions) -- a technical distinction, but worth considering the precision of the methodological claims. Also, while not a concern of the reviewers, discussion of theoretical implications and contextualization of the algorithmic approximations would increase overall understanding.

== Originality == While there has been work in selecting examples for ICL, the reviewers believed the specific approach to be novel and pragmatic (especially when considering algorithmic simplifications). This is a new area and there is the right balance of good conceptual motivation and heuristics to create a feasible implementation.

== Significance == As stated above, this is a increasingly widely-studied area (example selection in ICL) where small improvements can have large impact. Thus, there is definitely potential for impact, minimally as a baseline for future work. That being said, there were several concerns that the reviewers thought may hinder wide adoption including only using GPT-2 scale models (thus, without instruction-tuning and knowing that these results scale to SotA settings), only performing static selection (which is a natural area for future research with a multi-stage method), and questions regarding computational resources needed to perform this in practice. Thus, while the reviewers concur that the method is likely useful, it isn't clear what the impact will be with stronger LLMs (and if this ends up being a negative result, the impact will clearly be limited).